# Microstructures of FeCoNiMo and CrFeCoNiMo Alloys, and the Corrosion Properties in 1 M Nitric Acid and 1 M Sodium Chloride Solutions

**DOI:** 10.3390/ma15030888

**Published:** 2022-01-24

**Authors:** Chun-Huei Tsau, Meng-Chi Tsai, Wei-Li Wang

**Affiliations:** Institute of Nanomaterials, Chinese Culture University, Taipei 111, Taiwan; asd99586@yahoo.com.tw (M.-C.T.); willy11424@yahoo.com.tw (W.-L.W.)

**Keywords:** FeCoNiMo, CrFeCoNiMo, microstructures, hardness, potentiodynamic polarization curve

## Abstract

FeCoNiMo and CrFeCoNiMo equimolar alloys were prepared by arc-melting. The microstructures of the as-cast alloys were examined by SEM, HREM and XRD; and a potentiodynamic polarization test of the as-cast alloys was undertaken to evaluate the corrosion resistance in the solutions. Results showed that both of FeCoNiMo and CrFeCoNiMo equimolar alloys had a dendritic structure. The dendrites of these two alloys were a single phase which was a simple cubic (SC) structure with large lattice constant; and the interdendrities of these two alloys had a dual-phased eutectic structure in which the phases were face-centered cubic (FCC) and simple cubic (SC). The hardness of CrFeCoNiMo alloy was higher than that of FeCoNiMo alloy. Additionally, the potentiodynamic polarization test showed that CrFeCoNiMo alloy was better than FeCoNiMo alloy in 1 M nitric acid and 1 M sodium chloride solutions. Adding chromium into FeCoNiMo alloy could increase corrosion resistance in these two solutions. All of the results indicated that the CrFeCoNiMo alloy surpassed FeCoNiMo alloy.

## 1. Introduction

Nowadays, researchers can select the elements to prepare new materials for applications by following the high-entropy alloy concept [1,2,3]. The entropy of multicomponent alloy with solid solution states increases with increasing the number of elements in the alloys, and the alloys will be stable because Gibbs free energy decreases [4,5]. This concept is widely used to develop new alloys with good mechanical properties [6,7]. Additionally, the properties of high-entropy alloys can be improved by deformation [8,9]. Many alloys with good corrosion resistance are thus studied and developed, because corrosion is a major problem in the application of metals. Al_x_CrFe_1.5_MnNi_0.5_ alloy has good oxidation resistance at high temperature [10]. FeCoNiCrCu*_x_*, Al_7.5_Cr_22.5_Fe_35_Mn_20_Ni_15_ and Al_0.5_CoCrFeNi alloys all have good corrosion resistance in sodium chloride solutions [11,12,13]. Some of the alloys were developed as coating layers to improve corrosion resistance of base materials. Al_0.5_CoCrCuFeNi alloy is used as a coating alloy on the surface of AZ91D and thus improves the corrosion resistance in NaCl solution [14]. NbTiAlSiZrN_x_ alloy sputtered on the surface of 304 stainless steel showed good corrosion resistance in 1 N H_2_SO_4_ solution [15]. AlCr*_x_*NiCu_0.5_Mo alloys coated on the surface of Q235 steel can improve the corrosion resistance in NaCl solution [16]. Iron, nickel and chromium are the major elements of 304 stainless steel, and the fourth element, cobalt, is used to enhance the properties of 316 stainless steel. Therefore, many alloys are designed based on these four elements in order to achieve good corrosion resistance. Cu_0.5_NiAlCoCrFeSi alloy possessed lower corrosion current densities in H_2_SO_4_ and NaCl solutions when compared with 304 stainless steel [17]. Al_x_CoCrFeNi (x equal to 0.15 and 0.4) alloys showed better corrosion resistance when compared with HR3C steel at high temperature [18]. Adding small amount of Zr into AlFeCrCoNiZr_x_ alloys can improve the corrosion resistance in 0.5 M H_2_SO_4_ solution [19]. The addition of an appropriate amount of tantalum into CrFeCoNi alloy can increase the pitting resistance of CrFeCoNiTa_0.3_ alloy in 1 M sodium chloride solution [20]. Al_0.1_CoCrFeNi alloy showed higher polarization resistance and lower corrosion rate in a molten Na_2_SO_4_-NaCl eutectic mixture at 750 ± 5 °C when compared with nickel-based alloy 718 [21]. CoCrFeNiMo_0.85_ alloy used as a coating layer on stainless steel substrate could provide corrosion resistance in 3.5% NaCl solution [22].

FeCoNi and CrFeCoNi equimolar alloys showed a good corrosion resistance in 1 M sodium chloride solution as reported in our previous study [23]. Both FeCoNi and CrFeCoNi alloys had a FCC granular microstructure, and they were very soft. The average hardness of FeCoNi and CrFeCoNi alloys were 112 HV and 146 HV, respectively. The low hardness limits the application of FeCoNi and CrFeCoNi equimolar alloys. The addition of molybdenum can increase the corrosion resistance of stainless steels as is well known [24,25,26]. Adding molybdenum can also increase the hardness of FeCoNi and CrFeCoNi alloys because the radius of molybdenum atom is 1.40 Å, and the radiuses of Cr, Fe, Co and Ni are 1.28, 1.24, 1.25 and 1.25 Å, respectively [27]. Therefore, the present study investigated the effect of the addition of molybdenum on these alloys, and analyzed the microstructures, hardness and corrosion properties of FeCoNiMo and CrFeCoNiMo alloys.

## 2. Materials and Methods

The experimental equimolar alloys, FeCoNiMo and CrFeCoNiMo, were prepared by an arc furnace in an argon atmosphere. Each melt had a total weight of about 100 g. The nominal compositions are listed in Table 1. The as-cast alloys were examined in terms of the microstructures, hardness and corrosion properties. The microstructures of the alloys were observed by a scanning electron microscope (SEM, JEOL JSM-6335, JEOL Ltd., Tokyo, Japan) operated at 10 kV, and a high-resolution transmission electron microscope (HREM, JEOL JEM-3000F, JEOL Ltd., Tokyo, Japan) operated at 300 kV. The crystal structures of the phases in the alloys were examined by HREM diffraction patterns (DPs) which were transformed from the high-resolution lattice images by fast Fourier transformation (FFT) of Gatan digital micrograph software 1.4.1. The structures of the phases were also confirmed by an X-ray diffractometer (XRD, Rigaku, ME510-FM2, Rigaku Ltd., Tokyo, Japan) at a scanning rate of 0.04 degree/s. The overall hardness of the alloys was tested by a Matsuzawa Seiki MV1 Vicker’s hardness tester (Matsuzawa Co., Akita, Japan) under a load of 294 N (30 kgf); and the microhardness of the alloys was tested by a Mitutoyo Akashi MVK-G1500 (Mitutoyo Co., Kanagawa, Japan) microhardness tester under a load of 0.098 N (10 gf).

The potentiodynamic polarization curves of FeCoNiMo and CrFeCoNiMo alloys were measured by an electrochemical device (Autolab PGSTAT302N, Metrohm Autolab B.V., Utrecht, Netherlands) with three electrodes which were the specimen, a counter electrode and a reference electrode. The counter electrode was a platinum wire. The reference electrode was a saturated silver chloride electrode (Ag/AgCl). The reduction potential of a saturated silver chloride electrode (V_SSE_) is 0.2223 V higher than that of a standard hydrogen electrode (V_SHE_) [28]. The specimens were mounted by epoxy and had a fixed exposed area of 0.19635 cm^2^ (0.5 cm in diameter). All of the surfaces of the specimens were wet polished by #1200 SiC abrasive paper. The scanning rates of potentiodynamic polarization tests were all fixed at 1 mV/s. Nitrogen gas bubbling was used though the process to eliminate to effect of oxygen. The solutions were prepared by reagent-grade nitric acid, sodium chloride and distilled water.

## 3. Results and Discussion

Both FeCoNi and CrFeCoNi alloys had a granular microstructure, but CrFeCoNi alloy had a few Cr-rich HCP particles distributed in the matrix [23]. However, these two alloys, FeCoNiMo and CrFeCoNiMo, became dendritic microstructures after the addition of molybdenum. Figure 1a,b show the microstructures of FeCoNiMo and CrFeCoNiMo alloys, respectively. FeCoNiMo and CrFeCoNiMo alloys had a similar microstructure, the dendrites of these two alloys were simple cubic (SC) phase, and the interdendrities of these two alloys were a dual-phased (FCC and SC) eutectic microstructure. The chemical compositions of SC and FCC phases are listed in Table 2. The overall compositions of FeCoNiMo and CrFeCoNiMo alloys indicated those alloys matched the experimental design. The simple cubic phase (dendrites) of FeCoNiMo alloy had higher molybdenum content. The melting point of molybdenum is higher than the other elements, and thus the Mo-rich SC phase solidified first and formed the dendrites during solidification. The melting points of Cr, Fe, Co, Ni and Mo are 1875, 1536, 1498, 1453 and 2610 °C, respectively [27]. By contrast, the FCC phase in FeCoNiMo alloy had more Fe-, Co- and Ni-content. Similar results were also observed in CrFeCoNiMo alloy; simple cubic phase in this alloy had more Mo- and Cr-content because these two elements had higher melting points.

Figure 2 shows the TEM micrographs of FCC and SC phases in FeCoNiMo alloy. A simple FCC phase of FeCoNiMo alloy is shown in Figure 2a, and the inserts are the lattice image of FCC phase and corresponding FFT DP. The simple cubic phase in FeCoNiMo alloy also displays a pure morphology, as shown in Figure 2b; the inserts are the lattice image and corresponding FFT DP. The lattice image indicated that SC phase has a large lattice constant, and this was proved by the FFT DP. The small distances between the diffraction points indicated the large lattice constant of SC phase.

Figure 3 shows the TEM micrographs of FCC and SC phases in CrFeCoNiMo alloy. Figure 3a shows the morphology of FCC phase in CrFeCoNiMo alloy, and the insets are the lattice image of FFC phase and corresponding FFT DP. A few dislocations were observed in the FCC matrix; also the FFT DP indicated that it was a simple FCC phase. Figure 3b is the morphology of the SC phase in CrFeCoNiMo alloy, the inserts are the lattice image and corresponding FFT DP. The FFT DP also indicates that the distances between the diffraction points were very small, which meant a large lattice constant of the SC phase.

The XRD patterns of FeCoNiMo and CrFeCoNiMo alloys are shown in Figure 4. Only two phases, SC and FCC, in each alloy were analyzed from the XRD patterns. The lattice constants of SC and FCC phases in FeCoNiMo alloy were 8.25 and 3.54 Å, respectively. The lattice constants of SC and FCC phases in CrFeCoNiMo alloy were 8.40 and 3.58 Å, respectively. The results of XRD patterns matched the analysis by TEM. Therefore, adding molybdenum into FeCoNi and CrFeCoNi alloys forms the simple cubic phase with large lattice constant. Table 3 lists the overall hardness of the alloys, and the microhardness of the dendrites and interdendrities of each alloy. Results indicated that adding molybdenum could effectively increase the hardness of the alloys because it forms the second phase in the alloys and the solid solution strengthening effect. Molybdenum atoms have a large radius in comparison with the other elements. Therefore, adding molybdenum will induce the lattice distortion and result in solid solution strengthening.

The potentiodynamic polarization curves of FeCoNiMo and CrFeCoNiMo alloys tested in 1M deaerated HNO_3_ solution at 30 and 60 °C are shown in Figure 5a,b, respectively. Table 4 lists the potentiodynamic polarization data of the curves. The curve of each alloy with potential negative than corrosion potential (*E*_corr_) was the cathode which meant this alloy was under protected at this state. The cathodic current density of FeCoNiMo alloy was higher than that of CrFeCoNiMo alloy in 1 M deaerated HNO_3_ solution at 30 and 60 °C. The corrosion current density (*i*_corr_) of FeCoNiMo alloy was also higher than that of CrFeCoNiMo alloy. The curve of each alloy with potential positive than corrosion potential (*E*_corr_) was the anode which meant the alloy would be corroded at this state. The potentiodynamic polarization curve of CrFeCoNiMo alloy tested at 30 °C had a complete passivation region, as shown in Figure 5a; and almost no passivation potential (*E*_pp_) of this curve meant that it easily entered the passivation region. The passivation current density of CrFeCoNiMo alloy tested in 1 M deaerated HNO_3_ solution at 30 °C was almost the same. The passivation region of CrFeCoNiMo alloy under 30 °C broke down at potential (*E*_b_) of 1.14 V_SHE_ because of the oxygen evolution reaction [28]. By contrast, the potentiodynamic polarization curve of FeCoNiMo alloy tested at 30 °C showed a very poor corrosion resistance because of almost no passivation region. The potentiodynamic polarization curves of FeCoNiMo and CrFeCoNiMo alloys tested in 1 M deaerated HNO_3_ solution at 60 °C are shown in Figure 5b. Both of the *E*_corr_ and *i*_corr_ of FeCoNiMo and CrFeCoNiMo alloys increased while the test temperature increased to 60 °C. However, the passivation current density of CrFeCoNiMo alloy increased with increasing applied potential; and the passivation region of CrFeCoNiMo alloy under this condition was also broken down at 1.13 V_SHE_ because of the oxygen evolution reaction. However, FeCoNiMo alloy also displayed a poor corrosion resistance and no passivation region under this condition.

Figure 6 shows the micrographs of FeCoNiMo and CrFeCoNiMo alloys after the potentiodynamic polarization test in 1 M deaerated HNO_3_ solution at 30 and 60 °C. The morphology of FeCoNiMo alloy after the potentiodynamic polarization test at 30 °C displayed a uniform corrosion morphology, as shown in Figure 6a. However, the SC-phased dendrite was more corroded compared with the FCC phase. The dendrites of FeCoNiMo alloy were more severely corroded when the temperature increased to 60 °C. Figure 6b shows FeCoNiMo alloy after the potentiodynamic polarization test at 30 °C. The SC phase was almost corroded, the dendrites of FeCoNiMo alloy were all corroded and left dendrite-shaped cavities and FCC phase. Therefore, the SC phase of FeCoNiMo alloy had a very poor corrosion resistance in 1 M HNO_3_ solution. However, CrFeCoNiMo alloy showed a better corrosion resistance in 1 M HNO_3_ solution compared with FeCoNiMo alloy. Figure 6c,d are the micrographs of CrFeCoNiMo alloy after the potentiodynamic polarization test in 1 M deaerated HNO_3_ solution at 30 and 60 °C, respectively. These images show a uniform corrosion morphology, even when the temperature increased to 60 °C. However, the FCC phase of CrFeCoNiMo alloy corroded more in 1 M deaerated HNO_3_ solution. This was quite different with FeCoNiMo alloy, in which SC phase corroded more in 1 M deaerated HNO_3_ solution. Therefore, the corrosion resistance of CrFeCoNiMo alloy was better than that of FeCoNiMo alloy in 1 M HNO_3_ solution.

The potentiodynamic polarization curves of FeCoNiMo and CrFeCoNiMo alloys tested in 1 M deaerated NaCl solution at 30 and 60 °C are shown in Figure 7a,b, respectively. The potentiodynamic polarization data of FeCoNiMo and CrFeCoNiMo alloys tested in 1 M deaerated NaCl solution at 30 and 60 °C are listed in Table 5. The cathodic limit current density (*i*_L_) was observed in the cathodic curves of both FeCoNiMo and CrFeCoNiMo alloys. The *i*_L_ means the maximum reaction rate that is limited by the diffusion rate of hydroxyl ions (OH^-^) in solution [29]. FeCoNiMo alloy had larger *i*_L_ and *i*_corr_ compared with CrFeCoNiMo alloy; also *E*_corr_ of FeCoNiMo alloy was more positive than that of CrFeCoNi alloy in 1 M deaerated NaCl solution at 30 °C, as shown in Figure 7a. Both the potentiodynamic polarization curves of FeCoNiMo and CrFeCoNiMo alloys had anodic peaks and passivation regions in 1 M deaerated NaCl solution; but those of CrFeCoNiMo alloy were larger at 30 °C. The passivation potentials (*E*_pp_) and critical current densities (*i*_crit_) of anodic peaks of FeCoNiMo and CrFeCoNiMo alloys are listed in Table 5. Additionally, CrFeCoNiMo alloy had a broader passivation region and lower passivation current density (*i*_pass_) and a passivation region breakdown at 1.21 V_SHE_ because of oxygen evolution reaction. The passivation region of FeCoNiMo alloy was smaller and it broke down at 0.10 V_SHE_. Figure 7b shows the potentiodynamic polarization curves of FeCoNiMo and CrFeCoNiMo alloys in 1 M deaerated NaCl solution at 60 °C. In these two alloys, increasing test temperature in 1 M deaerated NaCl solution would increase the values of *i*_corr_ and *E*_corr_. Both *i*_crit_ and *i*_pass_ of CrFeCoNiMo alloys also increased with increasing temperature; the passivation region of CrFeCoNiMo alloy broke down at 1.17 V_SHE_. The shape of the passivation region of FeCoNiMo alloy changed when the temperature increased to 60 °C; and *i*_pass_ increased with increasing applied potential. The *E*_b_ of FeCoNiMo alloy at 60 °C was 0.17 V_SHE_.

Figure 8a,b show the surfaces of FeCoNiMo alloy after the potentiodynamic polarization test in 1 M deaerated NaCl solution at 30 and 60 °C, respectively. The surfaces were full of cracks and polygonal. The surfaces of FeCoNiMo alloy showed severe corrosion after the test; and the SC phase in FeCoNiMo alloy was highly corroded compared with the FCC phase. By contrast, the corrosion resistance of CrFeCoNiMo alloy in 1 M deaerated NaCl solution was better than that of FeCoNiMo alloy. Figure 8c,d show the surfaces of CrFeCoNiMo alloy after the potentiodynamic polarization test in 1 M deaerated NaCl solution at 30 and 60 °C, respectively. No significant difference was observed on the corroded surfaces of CrFeCoNiMo alloy tested at 30 and 60 °C. Also, no corrosion was found on the SC-phased dendrites of CrFeCoNiMo alloy. The major corrosion occurred on the FCC phase in the interdendrities. These micrographs indicate that CrFeCoNiMo had a better corrosion resistance than FeCoNiMo alloy in 1 M deaerated NaCl solution.

## 4. Conclusions

The present work studied the microstructures, hardness of FeCoNiMo and CrFeCoNiMo alloys, and the corrosion properties of these alloys in 1 M deaerated HNO_3_ and 1M deaerated NaCl solutions. Both FeCoNiMo and CrFeCoNiMo alloys had a dendritic microstructure; the dendrites of these alloys had a simple cubic (SC) structure, and the interdendrities had a dual-phased (FCC and SC) eutectic structure. Adding molybdenum into FeCoNi and CrFeCoNi alloys could effectively increase the hardness of the alloys, because they would form the second phase and display the effect of solid solution strengthening. The corrosion resistance of CrFeCoNiMo alloy was better than that of FeCoNiMo alloy in 1 M deaerated HNO_3_ and 1 M deaerated NaCl solutions. The SC phase of FeCoNiMo alloy corroded more in 1 M HNO_3_ and 1 M NaCl solutions. By contrast, the FCC phase of CrFeCoNiMo alloy corroded more in these two solutions.

## Figures and Tables

**Figure 1 materials-15-00888-f001:**
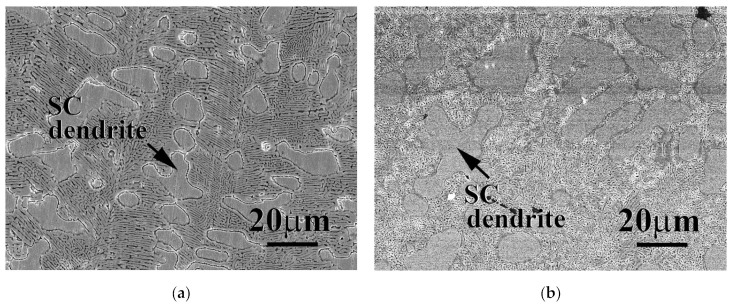
Scanning electron microscopy (SEM) micrographs of as-cast (**a**) FeCoNiMo alloy; and (**b**) CrFeCoNiMo alloy.

**Figure 2 materials-15-00888-f002:**
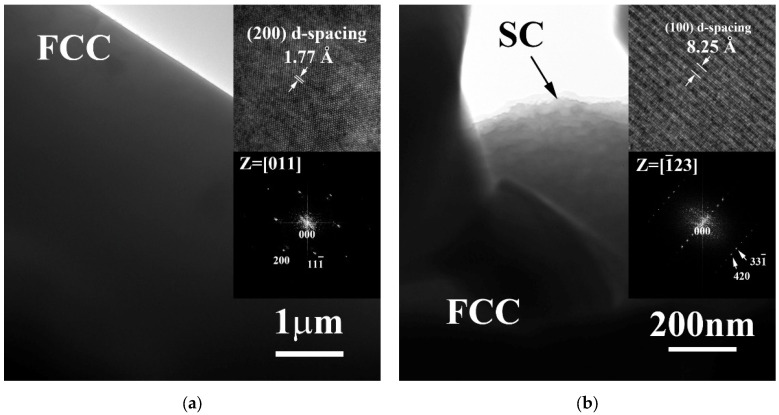
Transmission electron microscopy (TEM) micrographs of the phases in the as-cast FeCoNiMo alloy: (**a**) FCC phase, insets are the corresponding lattice image and fast Fourier transform (FFT) diffraction pattern, the image was taken from the zone axis of [011]; and (**b**) SC phase, insets are the corresponding lattice image and FFT diffraction pattern, and the image was taken from the zone axis of [$\bar{1}$23].

**Figure 3 materials-15-00888-f003:**
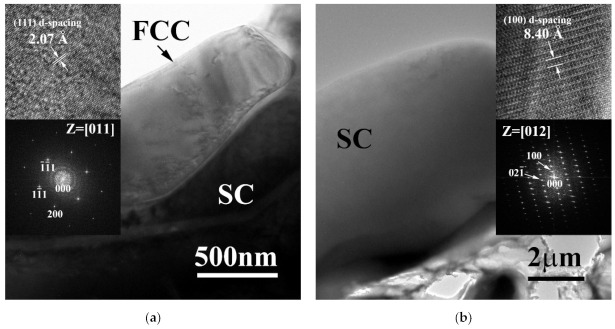
Transmission electron microscopy (TEM) micrographs of the phases in the as-cast CrFeCoNiMo alloy: (**a**) FCC phase, insets are the corresponding lattice image and FFT diffraction pattern, the image was taken from the zone axis of [011]; and (**b**) SC phase, insets are the corresponding lattice image and FFT diffraction pattern, the image was taken from the zone axis of [012].

**Figure 4 materials-15-00888-f004:**
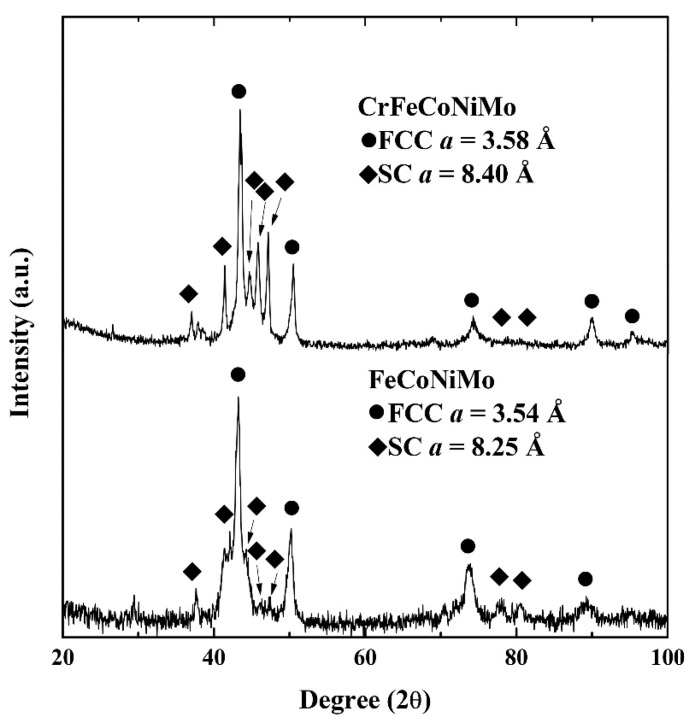
X-ray diffraction (XRD) patterns of as-cast FeCoNiMo and CrFeCoNiMo alloys.

**Figure 5 materials-15-00888-f005:**
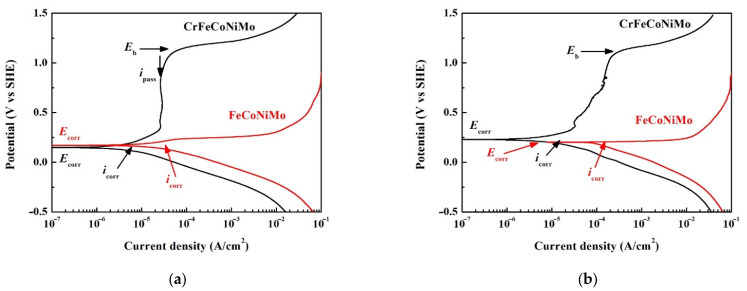
Potentiodynamic polarization curves of as-cast FeCoNiMo and CrFeCoNiMo alloys tested in 1 M deaerated HNO_3_ solution at (**a**) 30 °C; and (**b**) 60 °C. Red marks are for FeCoNiMo alloy and black marks are for CrFeCoNi alloy.

**Figure 6 materials-15-00888-f006:**
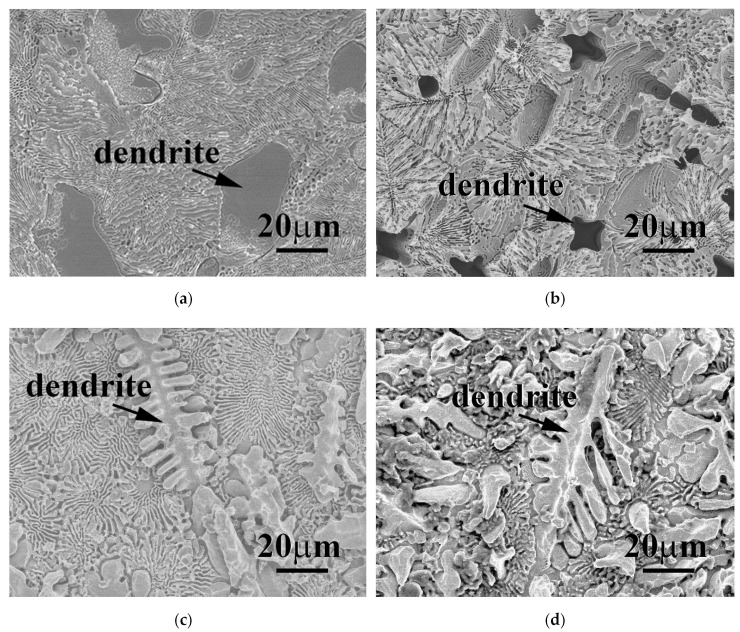
SEM micrographs of as-cast FeCoNiMo and CrFeCoNiMo alloys after potentiodynamic polarization tested in 1 M deaerated HNO_3_ solution at different temperatures: (**a**) FeCoNiMo alloy at 30 °C; (**b**) FeCoNiMo alloy at 60 °C; (**c**) CrFeCoNiMo alloy at 30 °C; and (**d**) CrFeCoNiMo alloy at 60 °C.

**Figure 7 materials-15-00888-f007:**
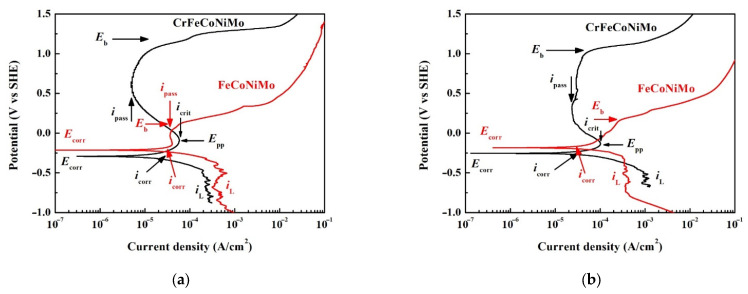
Potentiodynamic polarization curves of as-cast FeCoNiMo and CrFeCoNiMo alloys tested in 1 M deaerated NaCl solution at (**a**) 30 °C; and (**b**) 60 °C. Red marks are for FeCoNiMo alloy and black marks are for CrFeCoNi alloy.

**Figure 8 materials-15-00888-f008:**
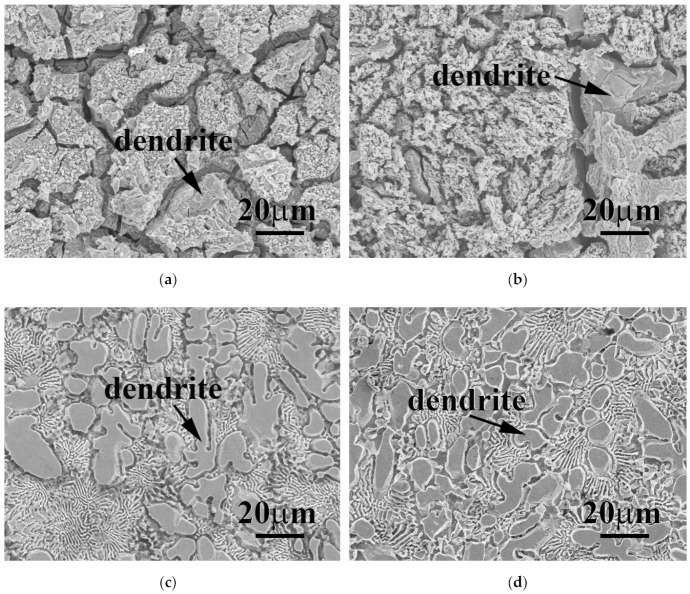
SEM micrographs of as-cast FeCoNiMo and CrFeCoNiMo alloys after potentiodynamic polarization tests in 1 M deaerated NaCl solution different temperatures: (**a**) FeCoNiMo alloy at 30 °C; (**b**) FeCoNiMo alloy at 60 °C; (**c**) CrFeCoNiMo alloy at 30 °C; and (**d**) CrFeCoNiMo alloy at 60 °C.

**Table 1 materials-15-00888-t001:** Nominal compositions of FeCoNiMo and CrFeCoNiMo alloys.

Alloy	Cr	Fe	Co	Ni	Mo
(at.%)	(wt.%)				
FeCoNiMo	N/A	20.73	21.87	21.79	35.61
CrFeCoNiMo	16.18	17.38	18.33	18.27	29.85

**Table 2 materials-15-00888-t002:** Chemical compositions of the phases in as-cast FeCoNiMo and CrFeCoNiMo alloy analyzed by SEM/energy-dispersive spectroscopy (EDS).

Alloy	Phase	Cr	Fe	Co	Ni	Mo
FeCoNiMo	overall	N/A	24.5 ± 0.2	24.2 ± 0.7	25.8 ± 0.2	25.4 ± 0.3
	FCC	N/A	26.3 ± 0.6	25.8 ± 0.3	26.0 ± 0.6	22.0 ± 0.2
	SC	N/A	23.4 ± 1.5	23.8 ± 1.0	22.0 ± 1.4	30.8 ± 3.9
CrFeCoNiMo	overall	20.1 ± 0.8	20.2 ± 0.2	19.5 ± 0.9	20.5 ± 0.3	19.7 ± 0.3
	FCC	19.2 ± 0.2	22.0 ± 0.6	21.3 ± 0.2	24.0 ± 1.4	13.5 ± 1.2
	SC	21.9 ± 1.0	17.7 ± 0.3	17.2 ± 0.7	14.4 ± 0.3	28.9 ± 0.5

**Table 3 materials-15-00888-t003:** Overall hardness of as-cast FeCoNiMo and CrFeCoNiMo alloys, and the microhardness of the dendrites and interdendrities in the alloys.

Alloy	Overall	Dendrites	Interdendrites
FeCoNiMo	471 ± 5 HV	585 ± 23 HV	315 ± 10 HV
CrFeCoNiMo	604 ± 8 HV	692 ± 18 HV	405 ± 9 HV

**Table 4 materials-15-00888-t004:** Potentiodynamic polarization data of as-cast FeCoNiMo and CrFeCoNiMo alloys tested in 1 M deaerated HNO_3_ solution at 30 and 60 °C.

Alloy	FeCoNiMo		CrFeCoNiMo	
Temperature (°C)	30	60	30	60
*i*_corr_ (μA/cm^2^)	36.0	153	6.0	14.7
*E*_corr_ (V_SHE_)	0.171	0.203	0.149	0.229
*i*_pass_ (μA/cm^2^)	N/A	N/A	30.0	N/A
*E*_b_ (V_SHE_)	N/A	N/A	1.14	1.13

**Table 5 materials-15-00888-t005:** Potentiodynamic polarization data of as-cast FeCoNiMo and CrFeCoNiMo alloys tested in 1 M deaerated NaCl solution at 30 and 60 °C.

Alloy	FeCoNiMo		CrFeCoNiMo	
Temperature (°C)	30	60	30	60
*i*_L_ (mA/cm^2^)	0.4	0.4	0.26	1.0
*i*_corr_ (μA/cm^2^)	31.0	61.0	15.0	60.0
*E*_corr_ (V_SHE_)	−0.214	−0.185	−0.292	−0.254
*E*_pp_ (V_SHE_)	−0.135	N/A	−0.089	−0.133
*i*_crit_ (μA/cm^2^)	40.4	N/A	58.3	102.0
*i*_pass_ (μA/cm^2^)	40.0	N/A	5.2	37.0
*E*_b_ (V_SHE_)	0.10	0.17	1.21	1.17

## Data Availability

Not applicable.

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
