# Peer review of "Microstructures of FeCoNiMo and CrFeCoNiMo Alloys, and the Corrosion Properties in 1 M Nitric Acid and 1 M Sodium Chloride Solutions"

_materials, 2022, doi:10.3390/ma15030888_

Round 1

Reviewer 1 Report

Authors did good study on the addition of Cr to improve the corrosion resistance. However, authors may address the following suggestions or comments to improve the quality of manuscript as well to make it more reader friendly.

1) It is well known fact that Cr addition will enhance the corrosion resistance; so what is the new in the present study?

2) Include highlights, graphical abstract with schematics

3) In section 2 : follow the standard format for representing the instruments or equipment used in the present study (Model, make, city, country).

4) Fig. 3 and 4: add the zone of axis.

Author Response

  • It is well known fact that Cr addition will enhance the corrosion resistance; so what is the new in the present study?

Reply: Our previous study on the FeCoNi and CrFeCoNi alloys (Mater. Chem. Phys. 2017, vol.186, 534-540) indicated that the FeCoNi alloys possessed better corrosion resistance than CrFeCoNi alloy. So we added the other elements into these two alloys; molybdenum was one of the elements. This manuscript studied the microstructures, hardness and corrosion behavior of these alloys. The poor corrosion resistance of FeCoNiMo alloy in 1M HNO3 solution was beyond our expectations. However, these results were also valuable for studying high-entropy alloys. Adding Cr to enhance the corrosion resistance of the alloy, maybe cause by the elements of Cr and Mo could be solid solute at temperature higher than 850 °C, and thus avoided to dissolution of Mo atoms. However, this needs more study.

2) Include highlights, graphical abstract with schematics

Reply: We used template to prepare our manuscript, but the template does not include these highlights, graphical abstract with schematics.

3) In section 2 : follow the standard format for representing the instruments or equipment used in the present study (Model, make, city, country).

Reply: They were added.

4) Fig. 3 and 4: add the zone of axis.

Reply: They have been added.

Reviewer 2 Report

Lines 140 to 145: “adding molybdenum could effectively increase the hardness of the alloys because of forming the second phase in the alloys and the solid solution strengthening effect…” on the effect of molybdenum on solid solution strengthening and second phase formation and increase in hardness, are in contradiction with Table 2 because FeCoNiMo alloy has more molybdenum than CrFeCoNiMo alloy but the lower hardness in all cases (Overall, dendrites, interdendrites). Please provide more information and evidence in this regard.
It is recommended to analyze corrosion products.
Electrochemical Impedance Spectroscopy (EIS) tests are recommended to examine the formation of corrosion products.
The literature review is not sufficient and authors must review and cite more papers in the field and especially newly published ones. Doing this, reviewing the following refs could be helpful:
Materials Science and Engineering: A, 825, 141875.
Journal of Alloys and Compounds 860, 158412
Measurement, 75, 2015, 5-11

Author Response

Lines 140 to 145: “adding molybdenum could effectively increase the hardness of the alloys because of forming the second phase in the alloys and the solid solution strengthening effect…” on the effect of molybdenum on solid solution strengthening and second phase formation and increase in hardness, are in contradiction with Table 2 because FeCoNiMo alloy has more molybdenum than CrFeCoNiMo alloy but the lower hardness in all cases (Overall, dendrites, interdendrites). Please provide more information and evidence in this regard.

Reply: FeCoNi and CrFeCoNi alloys were two different alloys. The hardness of FeCoNi and FeCoNiMo alloys were 112 HV and 471 HV, respectively; the increment of hardness was 420%. The hardness of CrFeCoNi and CrFeCoNiMo alloys were 146 HV and 604 HV, respectively; the increment of hardness was 414%. The increment of hardness of these two alloys were almost the same. Therefore, the description is correct.

It is recommended to analyze corrosion products.

Electrochemical Impedance Spectroscopy (EIS) tests are recommended to examine the formation of corrosion products.

Reply: Our equipment cannot measure the property of EIS. So, this is not included in the present study. However, we will do it in the future.

The literature review is not sufficient and authors must review and cite more papers in the field and especially newly published ones. Doing this, reviewing the following refs could be helpful:

Materials Science and Engineering: A, 825, 141875.

Journal of Alloys and Compounds 860, 158412

Measurement, 75, 2015, 5-11

Reply: We added 2 of the refs into our manuscript.

Reviewer 3 Report

In this paper, the authors studied microstructure and corrosion resistance of (Cr)FeCoNiMo HEAs, it is found that the CrFeCoNiMo alloy surpassed FeCoNiMo alloy.

Comments and suggestions:

1 In Fig.2(a), the lattice constant is labeled as 1.77Å,  while in Fig.4, it is indicated as 3.54Å,  they are not consistent; while 8.25 Å are the same. This needs further explanation; similar questions remain in Fig.3(a)  and Fig.4.

2 Horizontal label of Fig.4 needs to be revised, the item and unit reversed.

3 English needs to be improved overall.

Author Response

1 In Fig.2(a), the lattice constant is labeled as 1.77Å,  while in Fig.4, it is indicated as 3.54Å,  they are not consistent; while 8.25 Å are the same. This needs further explanation; similar questions remain in Fig.3(a)  and Fig.4.

Reply: Because FCC phase does not have (100) diffraction spot. We marked the d-spacing of (200) in Fig.2a (1.77 Å), and the d-spacing of (111) in Fig.3a (2.07 Å).

3 English needs to be improved overall.

Reply: We have rewritten our manuscript.

Round 2

Reviewer 2 Report

The revised manuscript is acceptable for publication.